# Physicochemical, Morphological, and Cytotoxic Properties of Brazilian Jackfruit (*Artocarpus heterophyllus*) Starch Scaffold Loaded with Silver Nanoparticles

**DOI:** 10.3390/jfb14030143

**Published:** 2023-03-03

**Authors:** José Filipe Bacalhau Rodrigues, Valeriano Soares Azevedo, Rebeca Peixoto Medeiros, Gislaine Bezerra de Carvalho Barreto, Maria Roberta de Oliveira Pinto, Marcus Vinicius Lia Fook, Maziar Montazerian

**Affiliations:** Academic Unit of Materials Science and Engineering, Federal University of Campina Grande, Campina Grande 58429-140, PB, Brazil; valeriano11@hotmail.com (V.S.A.); rebeca.peixoto@certbio.ufcg.edu.br (R.P.M.); gislainecarvalho11@hotmail.com (G.B.d.C.B.); roberta.oliveira@certbio.ufcg.edu.br (M.R.d.O.P.); marcus.liafook@certbio.ufcg.edu.br (M.V.L.F.)

**Keywords:** jackfruit, silver, nanoparticles, scaffold, biomaterials

## Abstract

Due to the physical, thermal, and biological properties of silver nanoparticles (AgNPs), as well as the biocompatibility and environmental safety of the naturally occurring polymeric component, polysaccharide-based composites containing AgNPs are a promising choice for the development of biomaterials. Starch is a low-cost, non-toxic, biocompatible, and tissue-healing natural polymer. The application of starch in various forms and its combination with metallic nanoparticles have contributed to the advancement of biomaterials. Few investigations into jackfruit starch with silver nanoparticle biocomposites exist. This research intends to explore the physicochemical, morphological, and cytotoxic properties of a Brazilian jackfruit starch-based scaffold loaded with AgNPs. The AgNPs were synthesized by chemical reduction and the scaffold was produced by gelatinization. X-ray diffraction (XRD), differential scanning calorimetry (DSC), scanning electron microscopy coupled with energy-dispersive spectroscopy (SEM-EDS), and Fourier-transform infrared spectroscopy (FTIR) were used to study the scaffold. The findings supported the development of stable, monodispersed, and triangular AgNPs. XRD and EDS analyses demonstrated the incorporation of silver nanoparticles. AgNPs could alter the scaffold’s crystallinity, roughness, and thermal stability without affecting its chemistry or physics. Triangular anisotropic AgNPs exhibited no toxicity against L929 cells at concentrations ranging from 6.25 × 10^−5^ to 1 × 10^−3^ mol·L^−1^, implying that the scaffolds might have had no adverse effects on the cells. The scaffolds prepared with jackfruit starch showed greater crystallinity and thermal stability, and absence of toxicity after the incorporation of triangular AgNPs. These findings indicate that jackfruit is a promising starch source for developing biomaterials.

## 1. Introduction

Scaffolds are three-dimensional, porous structures that facilitate tissue growth/remodeling by supporting human cell adhesion, proliferation, differentiation, and orientation in a stable environment. They are commonly employed in biomedical tissue engineering and are composed of biocompatible and biodegradable materials that allow for the incorporation and transport of medicines and biological components [1,2,3,4].

Scaffolds composed of metals, ceramics, polymers, and composites are used in biomedical tissue engineering [5]. Composite scaffolds are produced by mixing two or more materials, each with its own set of desirable properties. The tensile and compressive mechanical strengths of composite polymeric scaffolds, for example, are significantly higher than those of polymeric scaffolds [6]. As some are produced from biodegradable polymers, such as cellulose, chitosan, and starch, they are not only more durable, but also less toxic [6,7].

Starch is one of the main biopolymers present in nature [8]. It is an abundant, low-cost, biodegradable, biocompatible, and natural polymer widely used in fabricating scaffolds [9]. Amylose (glucose units joined by α-1,4-glycosidic bonds) and amylopectin (glucose units linked by glycosidic bonds at the α-1,4 and α-1,6 carbons) chains make up starch’s molecular structure [10]. Starch, like other polysaccharides and proteins, is a thermoplastic polymer composed of linear chains linked by weak bonds and is capable of being processed into membranes [11], gels [12], nanofibers [13], microparticles [14], nanoparticles [15], and scaffolds [16]. It can also encapsulate pharmaceuticals [16] and metallic nanoparticles [15] that bond to the long polymeric chains and hydroxyl groups of molecular structures.

Silver nanoparticles (AgNPs) stand out among metallic nanoparticles due to their distinctive physical, chemical, and biological features, particularly their anti-bacterial and antifungal activities [17], simple production, low cost [18], high conductivity, chemical stability, and catalytic activity [19]. They can be obtained with different morphologies, such as spheres, triangles, and rods; however, triangles are more biologically active [20,21].

AgNPs have a wide range of applications in biomedicine and are commonly coupled with biomaterials to provide bactericidal effects [22]. By attaching to peptidoglycans, AgNPs can permeate bacterial membranes, causing structural alterations, increased membrane permeability, and, eventually, death [23]. AgNPs can also interact with bacterial proteins through this mechanism, inhibiting DNA replication and bacterial growth [24].

AgNPs are used to improve the physical, chemical, and biological characteristics of biomaterials. A variety of studies have linked enhanced porosity, improved thermal and mechanical properties, increased water vapor absorption, anti-bacterial activity aggregation, and other biological aspects of biomaterials to the incorporation of AgNPs (see Table 1). Most of these studies used AgNPs with spherical forms due to their rapid synthesis and simplicity of acquisition [25] and employed more traditional polymer matrices, such as chitosan, collagen, cellulose, and alginate. Although the advanced properties of biomaterials loaded with spherical AgNPs produced from these biopolymers are well recognized, less is known about the impact that AgNPs of different morphologies can have on them, as well as their behavior in biomaterials produced from other polymer matrices, such as starch.

A few different types of starches can be utilized to produce scaffolds and biomaterials. They are derived from many plant sources and have varying compositions and properties that are determined by the plants’ growing circumstances, area, harvest season, and climate [26]. Commercial starches produced from wheat, corn, potato, rice, and cassava have already been extensively explored and are widely exploited in industry [27] as films [8], hydrogels [28], micro/nanoparticles [29,30], composites [31], and scaffolds of starches extracted from these sources. For this reason, searching for novel forms of local starch sources, especially in Brazil, would be critical, given their importance and the differences in their qualities.
jfb-14-00143-t001_Table 1Table 1Physical and biological properties of scaffolds, sponges, mats, and fibers loaded with silver nanoparticles prepared from starch and other polymers.MaterialAgNPs MorphologyPhysical PropertiesBiological PropertiesRef.Potato starch/PVA nanofibrous mats loaded with AgNPsSphericalHighly interconnected porous structure, relevant thermal stability, and fast release of AgNPsAbsence of toxicity against human fibroblast cells and AgNPs imparted anti-bacterial activity against *E. coli* and *S. aureus*[32]Carboxymethyl chitosan/oxidized starch sponges loaded with AgNPsSphericalImproved thermal propertiesBactericidal activity against *E. coli* and *S. aureus*; absence of toxicity and ability to promote the growth of L929 fibroblast cells; faster wound repair healing[33]Potato starch nanofibrous mats loaded with AgNPsSphericalReduced hydrophilicity and improved mechanical properties with increased tensile strength and reduced deformationBactericidal activity against *E. coli*; absence of toxicity against L929 fibroblasts cells up to AgNPs concentration of 2.5 mg.mL^-1^[34]Hybrid Ag nanoparticles/polyoxometalate–polydopamine nano-flowers Loaded chitosan/gelatin hydrogel scaffoldsSphericalHighly porous interconnected structureAnti-bacterial activity against *E. coli* and *S. aureus*; promoted wound healing; good biocompatibility with L929 cells and human umbilical vein endothelial cells (HUVEC)[35]Chitosan/Ag nanocomposite spongesSphericalHighly porous interconnected structure; improved mechanical properties and good water vapor transmission properties (in the range of 2000–2400 g/m^2^/d)Anti-bacterial activity against *E. coli* and *S. aureus*; non-cytotoxicity (cell viability values were greater than 90%)[36]Cellulose-based scaffolds containing orange essential oil and silver nanoparticlesSphericalUnexpected influence on water absorption and reduction in mechanical strength and elongation Anti-bacterial activity against *B. subtilis* and *E. coli*[37]3D cellulose nanofiber scaffolds decorated with silver nanoparticlesSphericalSilver nanoparticles significantly increased the thermal stability and mechanical properties, and reduced the scaffolds’ expansion process and swelling ratioAnti-bacterial activity against *E. coli* and *S. aureus* with 0.2 mM AgNPs content; low toxicity at concentrations of 0.2 and 1 mM AgNPs [38]3D hybrid scaffold based on collagen, chondroitin 4-sulfate, and fibronectin, functionalized with silver nanoparticlesSphericalMicrostructure with high number of interconnected pores*In vitro* cytocompatible and non-genotoxic in human gingival fibroblast cultures; biocompatibility with chick embryos CAM; antimicrobial activity against *F. nucleatum* and *P. gingivalis*[39]Poly(ε-caprolactone) nanocomposite fiber scaffolds loaded with silver nanowiresWiresIncreased crystallinity and thermal properties, and retarded enzymatic biodegradation Significantly enhanced cell proliferation of C2C12 mouse myoblast cells before and after electrical stimulations [40]Chitosan/*Bletilla striata*
polysaccharide composite scaffold loaded with silver nanoparticlesSphericalPreserved porous interconnected structure and improved mechanical propertiesEffective anti-bacterial and anti-biofilm properties both in vitro and in vivo against MRSA; potential to promote cell proliferation and angiogenesis[41]

Jackfruit (*Artocarpus heterophyllus*) provides a different type of starch, showing potential among starch sources. The high amylose concentration [26], low gelatinization temperature [42], and small gelatinization enthalpy change [43] of jackfruit starch make it a promising starch resource. However, jackfruit starch’s potential applications have not been fully studied by the scientific community; a limited number of studies have focused on this material, making more investigation into its potential role in the development of biomaterials imperative.

In this study, a jackfruit (*Artocarpus heterophyllus*) starch-based scaffold loaded with triangular AgNPs was synthesized and tested for its physicochemical, thermal, morphological, and cytotoxic properties. The production of such scaffolds utilizing readily accessible, inexpensive, and domestic starch sources was another primary objective of this study.

## 2. Materials and Methods

### 2.1. Chemicals

The jackfruit used for starch extraction was obtained from the local market of Campina Grande, Paraiba, Brazil. All chemical reagents were of analytical grade. Silver nitrate (AgNO_3_, purity 99.0%), tribasic sodium citrate dihydrate (Na_3_C_6_H_5_O_7_·2H_2_O, purity 99.0%), and hydrogen peroxide (H_2_O_2_, 35%) were purchased from Neon. Sodium borohydride (NaBH_4_, purity ≥96%) and glycerol (C_3_H_8_O_3_, purity ≥99%) were obtained from Sigma-Aldrich. The aqueous solutions were prepared with ultrapure water (18.2 mΩ·cm), obtained from a GEHAKA Master System MS2000 System. The L929 Mouse Fibroblast Cell Line (ATCC NCTC clone 929) was acquired from the Rio de Janeiro Cell Bank, Brazil.

### 2.2. Starch Extraction from Jackfruit Seed Endocarp

The starch extraction method was adopted from Perez, et al. [44]. In the first stage, jackfruit seeds were washed, peeled, and crushed in a blender until a uniform, thick, and homogenous mass was obtained, adding water in a 1:4 (m/v) ratio. The paste was filtered through organza bags (100 mesh). The filtered starch suspension was decanted for 24 h in a refrigerated atmosphere at 5 °C. The floating portion was removed, and the starch suspended in water was again decanted. This suspension and settling procedure was repeated until a white starch color was acquired. Following this, the starch was lyophilized (for 48 h) and sieved to 200 mesh.

### 2.3. Synthesis of AgNPs

AgNPs were synthesized through the chemical reduction method described by Zhang, et al. [45]. Initially, 30 mL of ultrapure water was transferred to a beaker and magnetic stirring (500 rpm) was performed at room temperature (25 °C). The system was then filled with 30 µL of silver nitrate (0.1 mol·L^−1^), 1.5 mL of sodium citrate (0.9 mmol·L^−1^), 60 µL of hydrogen peroxide (35%), and 200 µL of sodium borohydride (90 mmol·L^−1^). After adding sodium borohydride, the magnetic stirring was increased to 1150 rpm for 3 min, the period required for forming AgNPs. Our earlier study [46] provides more information on the effects of mixing intervals and variations in the volume and concentration of NaBH_4_ and H_2_O_2_ on the size, dispersion, and stability of AgNPs.

### 2.4. Synthesis of Jackfruit Starch Scaffold

The starch scaffold was obtained according to the methodology proposed by Perez, et al. [47]. Amounts of 7.5 g of starch and 2.5 mL of glycerol were first added to a beaker containing 250 mL of ultrapure water while magnetic stirring (300 rpm) was performed at 85 °C. The starch solution was then cooled to room temperature (25 °C), transferred to Petri dishes, and frozen (24 h), defrosted (2 h), frozen (24 h), and lyophilized (72 h).

### 2.5. Synthesis of Starch–AgNPs Scaffold

The starch–AgNPs scaffold was developed using the same process as the jackfruit starch scaffold, with a few modifications. After cooling to room temperature, 12.5 mL of the AgNP (a volume fraction of 5% to the starch solution) solution (1 × 10^−3^ mol·L^−1^) was added, followed by freezing (24 h), defrosting (2 h), freezing (24 h), and lyophilizing (72 h). As the AgNPs were homogenized in the starch solution, this resulted in a uniformly AgNP-decorated scaffold. Figure 1 illustrates the steps followed during the scaffold’s synthesis.

### 2.6. Characterization of the Silver Nanoparticles (AgNPs)

A UV-Vis spectrum model MB-102 (Bomem-Michelson, Vanier, Canada) was used to collect AgNPs spectra and confirm its synthesis. The scan was performed in the wavelength range of 250–1100 nm. The spectrums were collected using quartz cuvettes with a 10 mm optical path.

The size, polydispersity, and stability of AgNPs are properties that directly influence their biological properties [48]. They were determined using dynamic light scattering (DLS) and zeta potential (PZ) techniques. The analyses were carried out on a Brookhaven ZetaPals (Brookhaven Instruments, New York, USA). The tests were performed at room temperature without diluting the samples, with a scattering angle of 90°, laser wavelength of 632.8 nm (He-Ne), average viscosity of 0.887 mPa·s, and refractive index of 1.330. All measurements were taken three times.

The AgNPs’ morphology was investigated using a field-emission scanning electron microscope (FE-SEM) model S4700EI (Hitachi, Chiyoda, Japan) operating at a voltage of 15 kV. AgNPs were diluted in ultrapure water at a ratio of 1:10 and coated with platinum.

### 2.7. Scaffolds Characterization

When developing a biomaterial that contains metallic nanoparticles, it is very important to check that the particles have been incorporated. Therefore, X-ray diffraction (XRD) was used to prove the inclusion of AgNPs in the starch–AgNPs composite and to verify the crystallinity of the starch. The experiment was carried out on a X-ray diffractometer model XRD-7000 (Shimadzu, Kyoto, Japan) with CuKα radiation (1.5418), 40 kV, and 30 mA current in the interval of 10–80° and a resolution of 2°/min. The General System Analyzer Structure (GSAS II) program was utilized for Rietveld refinement.

The type of interaction that exists between a composite matrix and its filler provides information on how easily this filler will be released from the biomaterial. Thus, FTIR spectroscopy was employed to identify the vibration bands of starch and to assess the interaction between starch and AgNPs. The analysis was carried out on a Spectrum 400 FT-IR (Perkin Elmer, Waltham, MA, USA) device in the range of 4000–650 cm^−1^ with a resolution of 4 cm^−1^ in the diffuse reflectance mode for 32 scans at room temperature (25 °C) using an attenuated total reflectance (ATR) accessory equipped with a zinc selenium (ZnSe) crystal.

To analyze the existence of AgNPs in the scaffold, evaluate morphological changes following AgNPs incorporation, and identify the starch microstructure, scanning electron microscopy (SEM) was conducted on a SEM microscope model TM-1000 (Hitachi, Chiyoda, Japan) coupled with energy-dispersive X-ray spectroscopy (EDS) was performed. All images were taken from uncoated samples fixed on an aluminum alloy sample holder at a 15 kV accelerating voltage, 1 mm depth of focus, 30 nm resolution, low vacuum, and variable pressure (1 to 270 Pa). Under the same conditions, EDS analyses were performed on an EDS Quantax 50 XFlash (Bruker, Billerica, MA, USA). The EDS spectra were collected at 3 different points of the starch–AgNPs scaffolds to detect the AgNPs. Image processing was carried out using the Quantax 50 program.

The thermal stability of a biomaterial provides information on the conditions of use, storage, and application [49]. The thermal stability of the starch and starch–AgNPs scaffolds was evaluated using differential scanning calorimetry (DSC). A DSC, model 8500 (PerkinElmer, Waltham, USA), was employed in a temperature range of 25–300 °C with a heating rate of 10 °C/min under a nitrogen atmosphere and with a flow rate of 20 mL/min. an alumina crucible and a sample mass of 3.00 ± 0.05 mg were used.

### 2.8. Cytotoxicity Assessment

Cytotoxicity tests determine whether a biomaterial is safe or not [50]. The cytotoxicity assessment of the AgNPs, starch scaffold, and starch–AgNPs scaffold was performed according to the direct contact method described in ISO 10993-5:2009 [50] and ISO 10993-12:1998 [51]. Initially, 0.5 g of scaffold was weighed and sterilized for 30 min via UV radiation. Next, the extracts were prepared by submerging the scaffolds in 1.25 mL of PBS solution (0.05 M) for 24 h. Then, 100 µL/well of L929 cell suspension was seeded in 96-well plates and incubated for 24 h at 37 °C ± 1 °C under a 5% ± 1% CO_2_ atmosphere. After cultivation, 50 µL of the medium extracted from the samples was added and the cells were incubated for another 24 h.

For the AgNPs, five dilutions (1/1, 1/2, 1/4, 1/8, and 1/16) were performed to obtain final plaque concentrations ranging from 1 × 10^−3^ to 6.25 × 10^−5^ mol·mL^−1^. The method used for determining the concentration of AgNPs was solvent evaporation, which is well-known and widely used in the literature. In references [52,53], the researchers employed thermal gravimetric analysis to determine the weight of AgNPs and other metallic nanoparticles. Then, 100 µL/well of L929 cells suspension was seeded in 96-well plates and incubated for 24 h at 37 °C ± 1 °C under a 5% ± 1% CO_2_ atmosphere. Amounts of 20 µL of each of these dilutions were added to 96-well plates and the cells were incubated for another 24 h. The cytotoxicity assay was conducted following the MTT method.

After the incubation time, the culture medium was removed and 100 µL of MTT solution (5 mg·mL^−1^) was added; the plates were then incubated under the same conditions for another 4 h. The cells were then treated with 100 µL of DMSO to dissolve the formazan crystals. The plates were read following the optical density method on a Victor X3 microplate reader (PerkinElmer, Waltham, USA) at 570 nm with 650 nm reference filters. Latex sheets were used as the positive control and high-density polyethylene (HDPE) as the negative control.

## 3. Results and Discussion

### 3.1. Characterization of the Silver Nanoparticles (AgNPs)

The UV-Vis spectrum of the AgNPs colloidal solution is shown in Figure 2A. Three absorption bands were found in the UV-Vis analysis. According to the Schatz calculation, the first band at 333 nm corresponds to out-of-plane quadrupole resonance, the second shoulder-shaped band around 465 nm indicates dipole resonances characteristic of triangular nanoparticles [54], and the third band with maximum absorption at 744 nm resembles the plasmonic surface characteristic of the resonance band of almost perfectly triangular nanoparticles [54,55]. According to Mie’s hypothesis [56], anisotropic particles should be present because they have three absorption bands [55]. Furthermore, the colloidal solution of AgNPs acquired a blue color after synthesis, which can be attributed to the plasmonic excitation of the surface of triangular-shaped nanoparticles (nanoplates) [57].

The nanoparticles in the DLS result (Inset of Figure 2A) had an average size of 33.27 nm and a polydispersity index (PDI) of 0.205, which is typical of monodisperse solutions (0.300), which have values ranging from 0 to 1. As a result, the smaller the value, the more monodisperse the colloid [58]. The PZ findings revealed AgNPs with a surface charge of −33.39 mV, confirming stable AgNPs production [57]. The FE-SEM micrographs (Figure 2B–D) show the formation of nano-sized, anisotropic, and triangular monodisperse silver particles. These results are in agreement with those observed by others [59,60].

### 3.2. Morphological Characterization

SEM micrographs of the jackfruit starch granules, starch scaffold, and starch–AgNPs scaffold are shown in Figure 3A–F. The starch granules (Figure 3A,B) had a rounded and irregular bell shape ranging from 4–6 μm. Starch extracted from Brazilian jackfruit is typically of this size and form [61], whereas jackfruit starch extracted from Thai, Malaysian, or Chinese jackfruit is different in size and shape [26]. The variations in granule morphology are influenced by differences in cultivars and environmental factors [62] and affect starch’s functional characteristics [26]. The small grains and bell-shaped morphology of jackfruit starch developed for usage in the food and health industries contribute to its increased swelling capacity and viscosity.

The SEM micrographs of the starch scaffold and starch–AgNPs scaffold shown in Figure 3C–F revealed that both scaffolds had a porous structure created by well-defined pores arranged with high interconnectivity. Water vaporization is one possible mechanism that results in the formation of this porous structure. When water is removed, as in the lyophilization process, a large amount of porosity is created [63]. For the functional properties of starch, this same type of interconnected porous structure could also be observed in scaffolds prepared with starch extracted from rice [64], a source of starch that has a similar size and morphology to starch extracted from jackfruit [26]. This implies that the interconnected porous structure presented by the jackfruit starch scaffolds is influenced by the morphology and size of the starch grains extracted from the jackfruit.

For the starch–AgNPs scaffold (Figure 3E,F), it appears that adding AgNPs roughened the surface of the scaffold, facilitating cell attachment and proliferation [65,66]. Previous studies have reported a relationship between the incorporation of nanoparticles and increased total porosity of polymeric and ceramic composite scaffolds [67]. Herein, the incorporation of metallic nanoparticles into the starch polymer matrix was shown to modify the interactions between starch and glycerol [68,69], increasing its compactness and roughness.

The EDS analysis revealed the presence of AgNPs in the scaffold (Figure 3G). Starch was associated with levels of 59.4% carbon and 33.2% oxygen. The Fe content was linked to the sample holder, which was composed of an aluminum alloy with iron. The silver ions in the nanoparticles were responsible for the 2.0% silver content. The findings of Li, et al. [70] and Vaidhyanathan, et al. [71] are consistent with these results.

### 3.3. Phase Analysis

Figure 4 depicts the XRD pattern of the jackfruit starch, starch scaffold, and starch–AgNPs scaffold. Jackfruit starch’s XRD in Figure 4A shows diffraction peaks at 15°, 17°, 17.9°, 19.7°, 23.2°, and 23.7°, which are consistent with type-A crystalline structures and align closely with the data reported in the literature [72,73,74].

Three peaks (17°, 19.2°, and 23.6°) observed in the starch scaffold (Figure 4B) appeared at positions similar to those in the starch diffractogram (Figure 4A), but with amorphous characteristics and reduced intensity, indicating a partial loss of the type-A crystalline structure of jackfruit starch after processing (crystallinity = 16.95% by Rietveld refinement). Studies indicate that plasticized starch tends to form a V-type crystalline structure, which was indicated by the appearance of a slight shoulder around 17° and 19.2° (Figure 4B) [75]. In manufacturing starch scaffolds, glycerol was used, which acts as a plasticizer [76]. Therefore, it is suggested that, during processing, there was a change in the crystalline structure of starch through its interaction with glycerol.

After adding AgNPs to the scaffold (Figure 4C), the crystallinity increased to 33.88%, representing a 16.93% increase over the scaffold without AgNPs. It has been shown that the presence of AgNPs leads to a higher degree of crystallinity in polyimide films [77]. The starch–AgNPs scaffold exhibited four diffraction peaks at 37.80°, 44.02°, 64.36°, and 77.50°, which corresponded to planes (111), (200), (220), and (311) and correlated with the face-centered cubic structure of metallic silver (JCPDS file No. 03-0921). This validated the inclusion of AgNPs in the scaffold [78].

### 3.4. Scaffold Chemistry

The FTIR spectra of the starch, starch scaffold, and starch–AgNPs scaffold are shown in Figure 5. The vibration bands in the jackfruit starch spectrum (Figure 5A) represent features of the molecular deformations present in the starch molecules. The stretching deformation of –OH is responsible for the band located at 3400 cm^−1^ and angular deformation of the band at 1650 cm^−1^ [79]. The symmetrical stretching of –C–H is represented by the vibration band at 2926 and asymmetrical stretches at 2897 cm^−1^. The C–O–H bonds are represented by the bands at 1460–1400 cm^−1^ [80]. Absorptions at 1340 and 1024 cm^−1^ are due to –C–OH group deformations. Vibration modes associated with –C–C–H bonds were detected at 1418, 1205, and 1080 cm^−1^, whereas C–O, C–O–C, and C–C stretches corresponded to the 1153, 1107, and 933 cm^−1^ bands, which were typical of the vibration modes associated with pyranose rings found in natural polysaccharides [81]. All of these bands could be related to jackfruit starch bonds.

The FTIR spectra of the starch (Figure 5B) and starch–AgNPs (Figure 5C) scaffolds matched the jackfruit starch spectrum very well. Such behavior shows that the interaction of the AgNPs with the starch was physical, rather than chemical [82,83]. This suggests that, while AgNPs were present in the scaffold (as shown by the XRD and EDS data), they did not chemically interact with the starch’s polymer chains.

### 3.5. Thermal Analysis

Figure 6 depicts the thermograms of the starch and starch–AgNPs scaffolds, which display three endothermic peaks. The first referred to starch gelatinization, which had a gelatinization temperature of 81.3 °C in the starch scaffold and 71.3 °C in the starch–AgNPs scaffold [42,84]; the second peak, at temperatures of 110.2 °C and 105.8 °C in the starch scaffold and starch–AgNPs scaffold, could be attributed to the gelatinization of the amylose present in the starch, which was complexed by lipids [85,86]. The melting temperature (T_m_, third peak) of the starch increased from 146.3 °C to 185.2 °C with AgNPs incorporation. This peak was caused by the melting of crystalline starch domains that were reorganized during retrogradation [87].

The results revealed that, after incorporating AgNPs, the gelatinization temperatures of starch and amylose decreased while the T_m_ rose. There is evidence that AgNPs and other metallic nanoparticles have a greater effect on the T_m_ than they do on changing the initial degradation temperatures, as was the case for the gelatinization temperatures [83,88]. The increase in T_m_ could be explained by the fact that AgNPs are more thermally stable [83,89]. Shameli, et al. [90] reported similar results, observing an 18% increase in the enthalpy of starch–AgNPs composite films. The thermal stability was similarly enhanced when AgNPs were included in gelatin films [91] and sugar palm starch biocomposites [89].

### 3.6. Cytotoxicity Analysis

According to Figure 7A, it was observed that the tested dilutions of the AgNPs solution presented cytotoxicity values in the range of 99 to 86%. The highest dilution 1/16, corresponding to a concentration of 6.25 × 10^−5^ mol·L^−1^, showed a cell viability of 99%. By gradually increasing the concentration of AgNPs, a slow, but proportional, reduction in cell viability was noted. This was due to the AgNPs causing apoptosis and cell death [92]. However, even in the sample with the highest concentration of 1/1 (1 × 10^−3^ mol·L^−1^), a cell viability of 86% was observed, which, according to ISO 10993-5, indicates the absence of toxicity [50]. Therefore, triangular anisotropic AgNPs with a diameter of ~30 nm in the concentration range of 6.25 × 10^−5^ to 1 × 10^−3^ mol·L^−1^ were not toxic against L929 fibroblast cells.

Figure 7B demonstrates that the cell viability values of both scaffolds, one without AgNPs and one with, were 80.2% and 80.3%, respectively. Additionally, adding AgNPs did not result in any reduction in the viability of the cells, suggesting that AgNPs did not have any cytotoxic effects. The veracity of the data was confirmed by the fact that the negative control showed a hundred-percent cell proliferation. According to ISO 10993-5, if the material has a value that is lower than 70%, it is considered to have a toxic effect [50]. Both scaffolds exhibited cell survival values higher than 70%, demonstrating that the jackfruit starch scaffold loaded with AgNPs was not toxic to L929 cells, which suggests that it is appropriate for application as a biomaterial in treating various conditions, such as healing wounds.

## 4. Conclusions

In this study, we investigated the impact of incorporating silver nanoparticles (AgNPs) into a Brazilian jackfruit (*Artocarpus heterophyllus*) starch-based scaffold by analyzing its chemical, physical, thermal, and morphological characteristics. The chemical reduction technique yielded monodisperse, isotropic, and stable AgNPs with a size of 33.27 nm. The presence of AgNPs in the starch scaffold was verified by EDS and XRD analyses. The FTIR spectra of the jackfruit starch and starch–AgNPs scaffolds were identical, suggesting that the interaction of AgNPs with starch was purely physical. SEM revealed the scaffolds to have a very porous and three-dimensional structure. AgNPs inclusion maintained the scaffold’s porosity while increasing its surface roughness and crystallinity, which are all conducive to enhancing biological responses. In conclusion, the crystallinity, roughness, and melting temperature of the jackfruit starch scaffold were all improved by the addition of AgNPs. The scaffold could be developed using AgNPs to better survive temperature changes and demonstrate stronger biological interactions. The L929 cell survival rate was greater than 70% for AgNPs at concentrations of 6.25 × 10^−5^ to 1 × 10^−3^ mol·L^−1^ and for both scaffolds, confirming that triangular anisotropic AgNPs with a 30 nm diameter and the scaffold loaded with AgNPs were non-toxic to L929 cells and clinically relevant for treating wounds, for example. These findings help to fill a gap in our understanding of the toxicity and impacts of triangular anisotropic AgNPs on the physicochemical and biological properties of polymeric matrices. They have implications for a wide range of starch-based scaffoldings and highlight the potential use of jackfruit starch in biomaterial production. The next step would be using this approach to modify the biological and mechanical properties of scaffolds by including AgNPs.

## Figures and Tables

**Figure 1 jfb-14-00143-f001:**
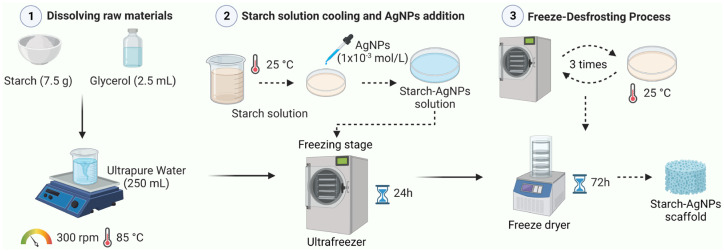
Schematic representation of starch–AgNPs scaffold’s synthesis.

**Figure 2 jfb-14-00143-f002:**
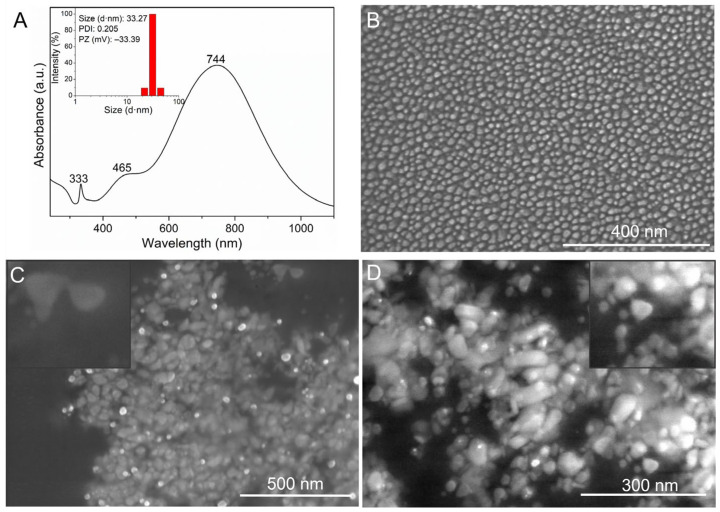
(**A**) UV-Vis spectrum of the AgNPs colloidal solution. Inset: Size distribution of AgNPs measured by the DLS. (**B**–**D**) FE-SEM micrograph showing the formation of nano-sized, anisotropic, triangular, and monodisperse silver particles.

**Figure 3 jfb-14-00143-f003:**
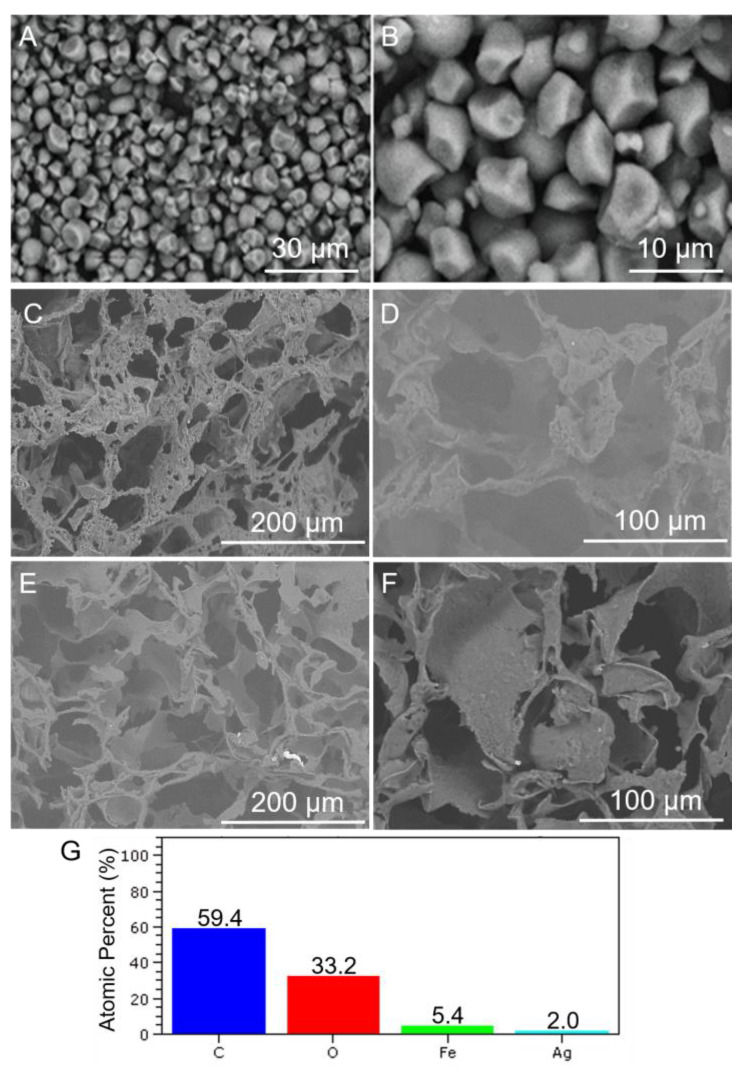
SEM micrographs of (**A**,**B**) jackfruit starch granules; (**C**,**D**) cross-sections of the starch scaffold and (**E**,**F**) starch–AgNPs scaffold, and (**G**) EDS analysis of the starch–AgNPs scaffold.

**Figure 4 jfb-14-00143-f004:**
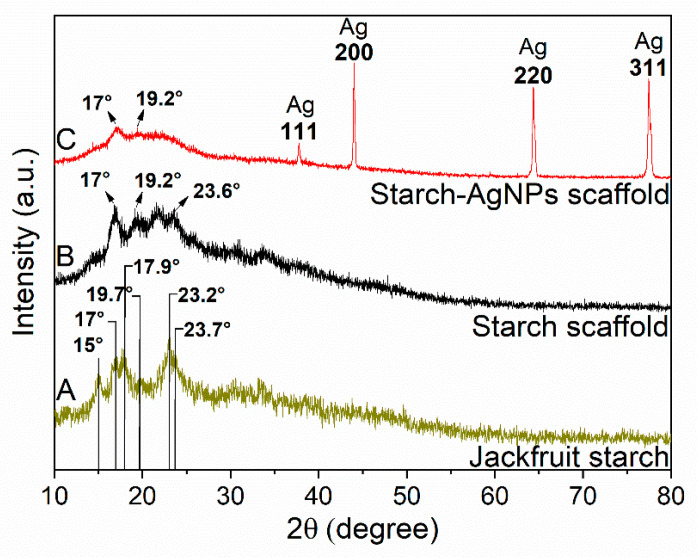
XRD patterns of the (**A**) jackfruit starch, (**B**) starch scaffold, and (**C**) starch–AgNPs scaffold.

**Figure 5 jfb-14-00143-f005:**
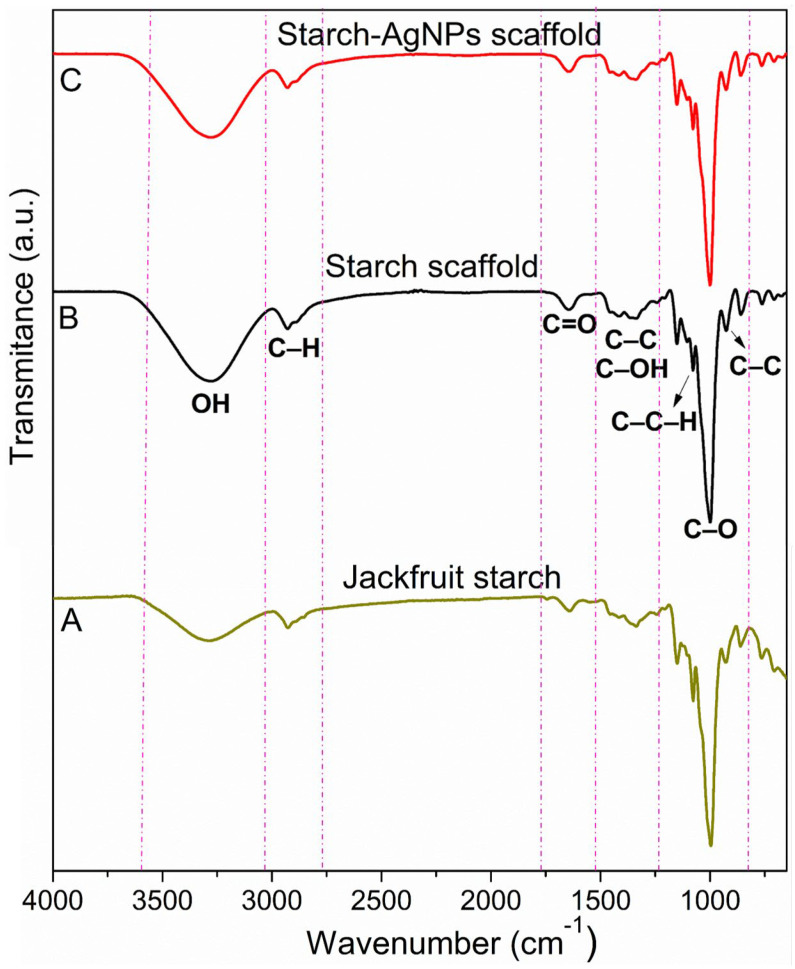
FTIR spectra of the (**A**) jackfruit starch, (**B**) starch scaffold, and (**C**) starch–AgNPs scaffold.

**Figure 6 jfb-14-00143-f006:**
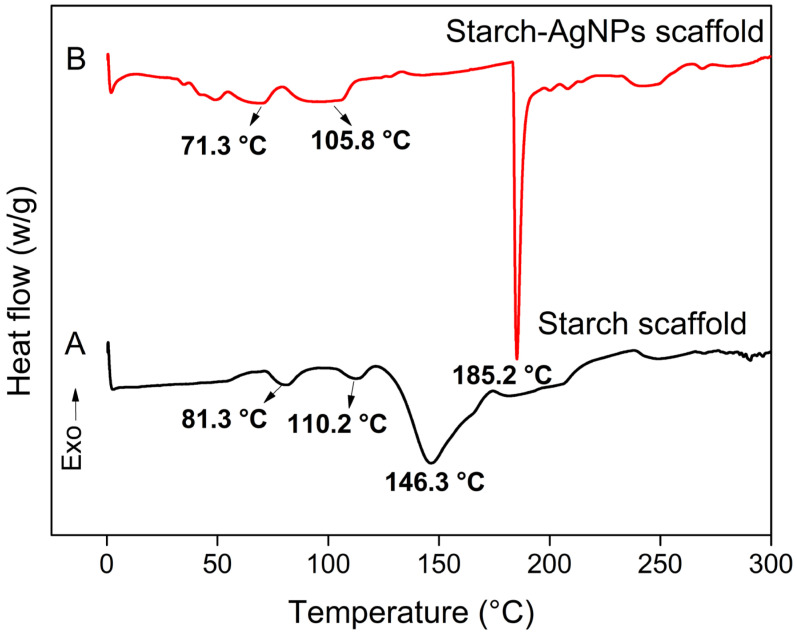
DSC analysis of (**A**) starch scaffold and (**B**) starch–AgNPs scaffold.

**Figure 7 jfb-14-00143-f007:**
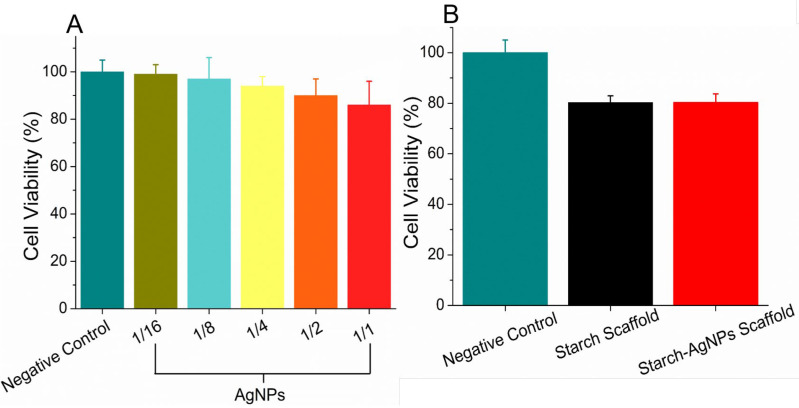
Cytotoxicity of AgNPs dilutions (**A**) and starch scaffold and starch–AgNPs scaffold (**B**) obtained by the MTT method.

## Data Availability

Not applicable.

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
