# Peer review of "Physicochemical, Morphological, and Cytotoxic Properties of Brazilian Jackfruit (Artocarpus heterophyllus) Starch Scaffold Loaded with Silver Nanoparticles"

_jfb, 2023, doi:10.3390/jfb14030143_

Round 1
Reviewer 1 Report
Dear Editor: Thank you for the opportunity to review the manuscript " Physicochemical, Morphological and Cytotoxic properties of 3 Brazilian Jackfruit (Artocarpus Eterophyllus) Starch Scaffold 4 loaded with Silver Nanoparticles" submitted to the Journal of Functional Biomaterials by Rodrigues et al. This manuscript (JFB-2022167) is prepared well, and I believe that the readers can have some benefits from this study. It is novel and a topic of interest to researchers in related areas. There are still some minor revisions that I suggest for its publication as seen below. 1) The introduction section is misleading. Include some recent articles related to starch/silver nanoparticles-based scaffolds materials. For example, DOI: 10.3390/molecules25163572, & https://doi.org/10.1007/s40204-019-00123-1. 2) Include a schematic diagram showing the Starch-AgNPs scaffold fabrication process. 3) Include some images of the cell culture studies. 4) How about the mechanical properties of the fabricated scaffolds? 5) Please indicate the presence of AgNPs on the SEM images i.e Fig. 2 (E & F). 6) What are the main barriers to the fabrication of starch-based scaffolds and how to address those? 7) There is some clumsy English grammar/phrasing throughout the manuscript. English/ grammar of the manuscript can be improved before publication.
Reviewer 2 Report
The manuscript entitled "Physicochemical, Morphological and Cytotoxic properties of Brazilian Jackfruit (Artocarpus Eterophyllus) Starch Scaffold loaded with Silver Nanoparticles" presents well-planned and well-performed, as well as extremely precisely described and interpreted research on the synthesis and very preliminary evaluation of the biological effect of Ag nanotriangles loaded into starch scaffold. I do not have serious comments on this articles, however, in my opinion the article is not of a sufficiently high scientific level for a journal with an impact factor> 4.5. Even if the section on the physicochemical characteristics of the material is at the appropriate level, the estimation of the biological effect by the MTT test is currently insufficient by the standards of any journal. There is lack of any other methods to visualize the biological effect, assess morphological changes, determine the type of cell death, etc. (fluorescence microscopy, SEM/TEM visualization, flow cytometry). In addition, the work lacks discussion of the results in relation to the current accessible scientific literature, now it is a report, not a scientific article. Finally, the work lacks emphasis on novelty, the emphasis should be on the morphology of Ag NPs due to the fact that there are not many studies investigating the biological effect of this morphology. Now, in this work it is completely left out and overlooked.
Reviewer 3 Report
This manuscript can be accepted for publication after the authors provide sufficient responses to the following comments:
1. Abstract, the authors mentioned "Starch is a widespread natural polymer used in healthcare applications due to its low cost and antibacterial properties", I doubt that pure starch has antibacterial property.
2. What is the novelty of the current study?
3. Section 2.2, what is homogeneous mass?
4. Section 2.5, the authors mentioned "...fruit scaffold, with a few modifications.", what is the modification conducted in the experiment?
Reviewer 4 Report
The manuscript entitled "Physicochemical, Morphological and Cytotoxic properties of Brazilian Jackfruit (Artocarpus Eterophyllus) Starch Scaffold loaded with Silver Nanoparticles" by Rodrigues et al. mainly prepared silver a nanoparticle-loaded Brazilian Jackfruit starch scaffold and then evaluated their physicochemical, morphological, and cytotoxic properties. The novelty and significance in this manuscript are slightly insufficient. Some experimental designs and detail information of original experimental data (please see questions) need to be added to make this manuscript better.
Some questions for the manuscript are as below.
1. Why did authors select a synthesized protocol of triangular Ag nanoparticles in this study? Did any difference was between triangular Ag nanoparticles and Ag nanoparticles (nanospheres) in this study? Authors should give some description in this manuscript.
2. Authors should provide TEM images of Ag nanoparticle-loaded starch scaffold and higher magnification of SEM image (Figure 1B). The morphology of triangular Ag nanoparticles could not be clearly observed.
3. Authors should provide higher magnification of SEM images of starch-Ag NPs scaffold (Figure 2E and 2F) to observe the distribution of Ag nanoparticles on starch scaffold.
4. Why was the amount of Fe more than Ag in Figure 2G? Where did Fe come from?
5. Authors mentioned the interaction between AgNPs and starch scaffold is physical adsorption. Were AgNPs easy to leave from the surface of starch scaffold during all further proceed steps (for examles, Cytotoxicity test)? Authors should provide this data.
6. Authors should provide the loading amount of AgNPs in starch scaffold.
7. In figure 6, what amount of AgNPs in starch-AgNPs scaffold? Current results in figure 6 could not absolutely indicate that starch-AgNPs scaffold (0.5 g) didn't have cytotoxicity for L929 cells. Because the amount of AgNPs was unknow. The used AgNPs amount was possibly lower than the threshold of Ag that could show cytotoxicity. The other question is that how and why authors decided to use the amount of 0.5 g starch-AgNPs scaffold in this study.
8. The manuscript title showed this study mainly focus on morphological and cytotoxic properties of starch-AgNPs scaffold. But both properties did not have detail discussion and experiments to evaluate them. I suggest that authors could provide more experiments and discussion in morphology (TEM images) and cytotoxicity (time- and dose-dependent cytotoxicity tests).
Reviewer 5 Report
Reviewer Reports:
I recommend a major amendment at this level.
General comments:
The manuscript entitled “Physicochemical, Morphological and Cytotoxic properties of Brazilian Jackfruit (Artocarpus Eterophyllus) Starch Scaffold loaded with Silver Nanoparticles” was reviewed. The work carried out in the manuscript is interesting and aimed at synthesizing a jackfruit (Artocarpus heterophyllus) starch-based scaffold loaded with AgNPs and testing its physicochemical, thermal, morphological, and cytotoxic properties for biomedical applications. However, the authors are suggested to undergo several substantial corrections as per the reviewer's comments to improve the quality of the manuscript. Better connect your research findings to previous works published in JFB and in other top journals. The innovation and the importance of this work are not clearly highlighted in the abstract, introduction and conclusions. The indication of this feature should start already from the abstract. Otherwise, not too many readers will bother reading the paper after looking at the abstract. Please work on this and prove to us why this work is valuable. Would you explicitly specify the novelty of your work? What progress against the most recent state-of-the-art similar studies was made? Additionally, the novelty of the research still is not clear and the discussion and conclusions can not satisfy me. Please also remove ANY lumped references. Please define each of them separately to avoid inappropriate citations. It is better to do not to use the first-person's pronoun. Do not use "we, us, or our" throughout the paper. It is recommended that the authors work with a science editor who is proficient in the Native English language to improve the organization and delivery of some portions of the manuscript. This will help improve the readability and help articulate better the relevance of the authors' work. The journal's author guidelines and instructions should be followed in preparing the revised version or resubmission. Other main remarks that in my opinion needs attention are the following:
Detailed comments:
Title: Ok.
Abstract:
The abstract should state briefly the purpose of the research, the principal results, and major conclusions. An abstract is often presented separately from the article, so it must be able to stand alone. In the abstract, please add an indication of the achievements from your study that are relevant to the journal's scope. Please be concise - maximum 1-2 lines.
Introduction:
The review of the literature needs more updating with works to have a clear and concise state-of-the-art analysis. This should more clearly show the knowledge gaps identified and link them to the paper goals. The introduction section is poorly organized. While the general introduction is acceptable, the state-of-the-art review that follows is very difficult to understand and no specific thoughts can be inferred. Please provide a table and compare current work with others. The major defect of this study is the debate or argument is not clearly stated. In addition; the introduction should be clearly stated the research questions and targets first. Then answer several questions: Why is the topic important (or why do you study it)? What are the research questions? What has been studied? What are your contributions? Why is it to propose this particular method?
The relevant reference may be of interest to the author according to below:https://link.springer.com/article/10.1007/s12649-020-01083-5
https://www.mdpi.com/2071-1050/14/19/12942
https://www.sciencedirect.com/science/article/abs/pii/S0013935121015796
Please eliminate the use of redundant words. Eg. In this way, Recently, Respectively, therefore, currently, thus, hence, finally, to do this, first, in order, however, moreover, nowadays, today, consequently, in addition, additionally, furthermore. Please revise all similar cases, as removing these term(s) would not significantly affect the meaning of the sentence. This will keep the manuscript as CONCISE as possible. Please check ALL. Avoid beginning or ending a sentence with one or a few words, they are usually redundant. Kindly revise all.
Materials and Methods:
Please avoid having one heading after another with no discussion in between as in the case of Sections 2. and 2.1 Kindly inspect the entire document for similar instances and revise accordingly. Please add in the beginning your scientific hypothesis. In the course of describing the performed actions, please provide reader guidance, sufficient for understanding why those actions have been performed. The percentage purity and company of all reagents/chemicals utilized must be reported. Though some of the model/brands of the equipment used was stated, their country of manufacture should be reported as well.
Results and Discussion:
All the findings of the current work need to be compared and discussed with the results of other researchers finding instead of having a general comparison with other researchers' works. The authors should perform a comparison between the forecasting results. In your discussion section, please link your empirical results with a broader and deeper literature review.
Conclusions:
Please make sure your conclusions section underscores the scientific value-added of your paper, and/or the applicability of your findings/results. Highlights the novelty of your study.
References:
Please check the reference section carefully and correct the inconsistency.
Round 2
Reviewer 2 Report
I do not find the response to my comments as satisfactory.
Moreover, how the authors calculated the concentration of Ag NPs in mol/L (Triangular anisotropic AgNps revealed no toxicity against L929 cells at concentrations ranging from 6.25×10-5 a 1×10-3 mol.L-1). It is difficult to calculate the concentration of trangular Ag NPs and there is no information about the method of this precise data determined for this morphology. In the added table there is no examples of Ag nanotriangles, so this is not discussion on the other authors' results.
I do not recommend this paper for publication.
I leave the decision to the editor if other reviewers agree for acceptance of this paper.
Reviewer 4 Report
Authors had addressed all questions. I recommend this manuscript in current form could be accepted by Journal of Functional Biomaterials.
Author Response
Thank you!
Reviewer 5 Report
Reviewer Reports:
I have reviewed the revised version of the manuscript entitled “Physicochemical, Morphological and Cytotoxic properties of Brazilian Jackfruit (Artocarpus Eterophyllus) Starch Scaffold loaded with Silver Nanoparticles”. The authors have carefully addressed and explained the comments. The manuscript is ok in this format.
Author Response
Thank you!
Round 3
Reviewer 2 Report
I do not agree with the authours. We can not estimate concentration of Ag NPs in mol/L. NPs are not one a simple atom of Ag but the cluster of atoms. The authors do not know their exact number of Ag atoms in a particular nanocrystal. So we can not calculate the concetration of NPs in the same way as in the case of solutions of salts.
Of course, it is possibility to calculate concentration of NPs in the case of small spherical NPs knowing their avaerage radium and molar absorption coefficient. In teh case of trangular morphology it is more complicated.
I am not conviced by the the papers that was given by authors as examples of using this approach because I do not agree with this. I checked one of them and the authirs determined concentration of Ag NPs covered by DNA and give this concentration in ng/ml. It is absolutetly something different.
The only correct way to determined "concentration" of your Ag NPs is to express it in mg/ml. I mean the dry weight of the sample in a given volume.
I leave the final decision to the editor, but I do not accept the work for publication because it contains serious substantive incorrections